# Hypergravity as a gravitational therapy mitigates the effects of knee osteoarthritis on the musculoskeletal system in a murine model

Benoit Dechaumet[1], Damien Cleret[1], Marie-Thérèse Linossier[1], Arnaud Vanden-Bossche[1], Stéphanie Chanon[2], Etienne Lefai[2], Norbert Laroche[1], Marie-Hélène Lafage-Proust[1], Laurence Vico[1]*

1 SAINBIOSE Laboratory, INSERM, University of Lyon, Saint-Etienne, France, 2 CarMeN Laboratory, INSERM, INRA, University of Lyon, Pierre-Bénite, France

☯ These authors contributed equally to this work.
* vico@univ-st-etienne.fr

## Abstract

Insights into the effects of osteoarthritis (OA) and physical interventions on the musculoskeletal system are limited. Our goal was to analyze musculoskeletal changes in OA mice and test the efficacy of 8-week exposure to hypergravity, as a replacement of physical activity. 16-week-old male (C57BL/6J) mice allocated to sham control and OA groups not centrifuged (Ctrl 1g and OA 1g, respectively) or centrifuged at 2g acceleration (Ctrl 2g and OA 2g). OA 1g displayed decreased trabecular bone in the proximal tibia metaphysis and increased osteoclastic activity and local TNFα gene expression, all entirely prevented by 2g gravitational therapy. However, while cortical bone of tibia midshaft was preserved in OA 1g (vs. ctrl), it is thinner in OA 2g (vs. OA 1g). In the hind limb, OA at 1g increased fibers with lipid droplets by 48% in the tibialis anterior, a fact fully prevented by 2g. In Ctrl, 2g increased soleus, tibialis anterior and gastrocnemius masses. In the soleus of both Ctrl and OA, 2g induced larger fibers and a switch from type-II to type-I fiber. Catabolic (myostatin and its receptor activin RIIb and visfatine) and anabolic (FNDC5) genes dramatically increased in Ctrl 2g and OA 2g (p<0.01 vs 1g). Nevertheless, the overexpression of FNDC5 (and follistatine) was smaller in OA 2g than in Ctrl 2g. Thus, hypergravity in OA mice produced positive effects for trabecular bone and muscle typology, similar to resistance exercises, but negative effects for cortical bone.

## Introduction

The hallmark of osteoarthritis (OA) is loss of articular cartilage, subchondral bone disturbances such as sclerotic changes and osteophyte growth [1, 2]. OA is a progressive joint disease, mainly afflicting the weight-bearing knee joint of older adults, with more than 200 million people affected worldwide [1, 3]. OA patients have pain related to joint degeneration, inflammation and joint stiffness that reduce range of motion [4–7]. These disabilities can combine with depressive states and expose them to a negative spiral of accumulation of negative effects due to sedentary lifestyle

**Data Availability Statement:** All relevant data are within the manuscript and its Supporting information files.

**Funding:** LV: CNES (Centre National d'Etudes Spatiales, contract n°4800000899) https://www.cnes.fr/ LV and MHLP: ANR-10-EQPX-06-01 https://anr.fr/ProjetIA-10-EQPX-0006 The funders had no role in study design, data collection and analysis, decision to publish, or preparation of the manuscript.

**Competing interests:** The authors have declared that no competing interests exist.

[8–11]. Such disuse atrophy of muscle is associated with ectopic adipogenesis and loss of knee muscle strength with limited flexion range [4]. Relationships between OA and osteoporosis have also been widely discussed [12]. Some studies [13, 14] showed that OA is associated with higher bone mineral density [BMD]. However, there is also evidence suggesting that high BMD level is protective for OA [15]. Recently, it has been shown, in patients scheduled for knee arthroplasty, that those who develop severe OA and attrition have lower BMD [16]. Similarly, in C57BL/6 mice who underwent surgical-induced knee OA, bone loss in the femoral epiphysis and metaphysis was observed 28 to 30 weeks post-surgery through X-ray micro-computed tomography [µCT] and histology [2].

Overall, these studies have suggested that sarcopenia and osteopenia/osteoporosis—or osteo-sarcopenia—are two conditions that might be associated with OA. Bone and muscle influence each other at a mechanical, chemical and metabolic level. Skeletal muscle and bone marrow displayed fatty infiltration with age, a phenomenon amplified by sarcopenia and osteoporosis [17]. Osteosarcopenic subjects are more prone to falls and therefore to fractures, which leads to personal and societal costs [17]. Studying possible pathological changes in musculoskeletal system of OA patients is urgently required to improve exercise therapy and to develop appropriate rehabilitation programs.

With this in mind, our first goal was to analyze bone and skeletal muscle in OA mice using the model of destabilization of the medial meniscus (DMM) as it has been shown to resemble closely to slowly progressive human OA [18].

Treatments for knee OA attempt to reduce pain, joint stiffness, inflammation and to increase range of motion through a variety of approaches. These approaches include pharmacological, surgical, physical therapy modalities, orthotics & braces and exercise. Neuromuscular electrical stimulation or volitional contraction alone, or in combination have been tested with mixed results [19–21]. Moderate or high-load exercises specifically designed to strengthen muscles have been the focus of most rehabilitation exercises [22]. In some studies, resistive exercises including programs to strengthen knee extension and flexion have resulted in significant strength gains and appear to reduce pain and improve function [23]. In others, the American College of Sports Medicine's Therapeutic Strengthening Programs do strengthen knee extensors, but do not reduce pain or disability [24]. Such programs might not be well tolerated because of the torque exerted about the knee joint [25].

Osteosarcopenia can start a vicious circle of weakness and inactivity and favor overweight and obesity, which are strong factors for the onset of knee OA [26]. Obesity induces excess load and hyperglycemia-related metabolic disturbances, both acting on joint function. However, the respective role of each factor is not well known. On the one hand, walking while carrying additional weight has been suggested as practical intervention for improving muscle strength and bone quality [27, 28]. On the other hand, a recent meta-analysis concluded that type 2 diabetes is associated with OA development even when controlling for body weight [29], suggesting that metabolic syndrome is prevalent over body weight.

Our second goal was to test a new intervention, whereby OA mice were exposed to chronic hypergravity using a centrifuge force added to the earth gravity. We postulated that hypergravity could mimic physical exercise by increasing body weight (not body mass), while avoiding exercise-associated physical barriers. This is done by housing animals in a carousel whose gondolas are gimballed in such a way that the resultant force is perpendicular to the cage floor [30]. In healthy mice, chronic hypergravity acted as endurance training on muscle force until 3g [30] and exhibited beneficial bone effects at 2g [31]. As a result, we performed a 2g intervention to test whether it could slow down OA progression and keep the musculoskeletal system healthy. Intervention was applied shortly after OA induction, as it is believed that early knee OA is the window of opportunity for interventions that benefit the patient [32]. The other

outcome was to specify the etiology of OA in obesity regarding the role of overload respective to metabolic factors, since 2g mice mimic a doubled body weight but not inflammatory and metabolic profiles of obesity.

In the present study, we examined the effects of 8-week exposure at 2g on bone and skeletal muscle of intact and OA mice on the right DMM limb. OA was scored, bone microstructure was investigated at the tibial trabecular and cortical compartment levels using nano computed tomography (nanoCT). In addition, bone cellular activities were measured after mineralizing fronts and Tartrate-Resistant Acid Phosphatase (TRAP) staining and calf muscle morphology and fiber-type switch was determined. We also investigated ectopic fat invasion in musculo-skeletal tissues and cytokines and humoral factors linking muscle to bone. Elevated blood vascularity at the osteochondral junction of osteoarthritic knees has been reported as one of the several factors associated with initiation and progression of osteoarthritis [33]. Lastly, we also investigated whether vascular density was challenged in bone and muscles.

## Material and methods

### Animals

Sixty-four C57BL/6J male mice of sixteen-week-old (Charles River Laboratories, L'Arbresle, France) were used in this study. During the experiment, 2 to 3 mice in standard cages were housed in the room hosting the centrifuge (see description below) at (22˚C), 50% relative humidity with a 12/12 h light-dark cycle throughout the study. Food and water were provided ad libitum (Safe diets A04, Augy, France). At 24-weeks, they were euthanized by cervical dislocation. Protocols and animal procedures conformed to the European community standards of care used on laboratory animals (Ministère de l'Agriculture, France, Authorization No. 42-21-080) and were approved under numbers CU14N06 and CU15N06 by the local animal care and UJM committee (Comité d'Ethique en Expérimentation Animale de la Loire -Université Jean Monnet, CEEAL-UJM).

### Surgical induction of osteoarthritis

The surgical OA model was induced in thirty-two 16-week-old mice by scalpel incision of the cranial menisco-tibial ligament of the medial meniscus in the right knee joint of the hind limb as described previously [34]. The other 32 mice underwent the same surgery except that the menisco-tibial ligament was not sectioned and serve as sham-operated controls (Ctrl). During surgery, all animals were anesthetized by inhalation of isoflurane. Mice were allowed free cage activity after surgery. No mortality occurred. To minimize suffering peri- and post-operatively, meloxicam (0.5mg/ml) a non-steroidal anti-inflammatory drug, was given in drinking water 48h before and 48h after surgery.

### Hypergravity exposure

Two days after the induction of OA, the 8-week hypergravity experiment began. One Ctrl and one OA group remained in the centrifuge room (Ctrl 1g and OA 1g, respectively) while, the centrifuged mice (Ctrl 2g and OA 2g) were transferred into the gondolas (56.2 cm x 52 cm x 59.2 cm) of the centrifuge (radius 1.4 m) on the hypergravity platform. The centrifuge (COMAT Aérospace, Flourens, France) made it possible to maintain a permanent level of hypergravity. The centrifuge with a radius of 1.4 m had four gondolas hanging on the periphery. Each gondola could accommodate up to four cages. All gondolas were equipped with a video surveillance system to control animals' condition and food/water stocks. Based on previous work [31], an acceleration of 2g (rotation speed of 29.6 rotations per minute) was fixed over the hypergravity period of 8 weeks

(S1 Video and S1 Fig in S1 File). The duration of both spin-up and spin-down is 40 sec. Animals were provided with sufficient food and water for 21–24 days. The centrifuge was stopped at three intervals: at mid-experiment for maintenance, 7 days before the end of experiment for maintenance and first injection of tetracycline fluorochrome for dynamic histomorphometry (see below) and lastly 2 days before the end of experiment for the second tetracycline injection.

## Vascular infusion and nano-computed tomography (nano-CT)

Ten out of 16 euthanized mice per group were infused via the left ventricle with a barium sulfate solution, an X-ray contrasting agent, to fill up blood vessel according to a method validated in our team [35]. Following the infusion, the right knee was obtained by cutting half the femur and tibia/fibula above and below the joint line. Excess muscles were removed, and joints were fixed in 4% PFA (48h at 4˚C), embedded undecalcified with knee in extension in methyl-methacrylate (MMA) resin for nano-CT and histology analysis.

A standardized parallel-beam nano-CT scan was performed within a field of view of 7.2mm height and 5mm width using a nano-CT (GE Phoenix Nanotom m, Wunstorf, Germany). Scans were conducted at 90 kV, 120 µA with a 750 ms exposure time and a rotation step of 0.15˚. The source-to-sample distance was 15 mm and the source-to-detector distance was 250 mm, leading to an effective pixel size of 3 µm. The scanned region included all tissues between distal femoral epiphysis up to proximal tibia including the metaphysis. An XY image (1307 $^*$ 1228 pixels) stack was obtained. As seen in Fig 1a and 1c, structures appearing in bright white represent the vascular network due to the high contrasting power of barium sulfate, while the bone appeared gray and the marrow in black (non-contrasted).

## Tibia bone parameters

ImageJ was used to quantify distinctive tibia parameters on nano-CT images. Transformation of the sagittal section stack into a frontal section stack using ImageJ was required before the software could quantify the 2D and 3D bone and vascular parameters as seen in Fig 1. At the diaphysis, 3mm under the growth plate cartilage on a 1.2mm thick ring, cortical porosity (Ct.Po, %) and cortical thickness (Ct.Th, µm) were evaluated (Fig 1c). At proximal trabecular metaphysis, bone and marrow vascular parameters (Fig 1a and 1b) were analyzed with CTAn software (CTAnalyser®; SkyScan). They include bone volume (BV/TV, %), trabecular separation (Tb Sp, µm), trabecular number (Tb N, number) and blood vessel density (vessel number/200 µm$^2$).

After nano-CT analysis, the samples were processed for dynamic histomorphometry. The proximal tibia was cut (LEICA SM 2500E microtome) in 9µm-thick serial frontal sections. Measurements were done semiautomatically with digitizing tablet (Summasketch, Summagraphics, Paris, France) and a software designed in our laboratory [31]. Knee OA was evaluated on the medial compartment of the right knee joint. At this level, we measured epiphyseal articular cartilage thickness (µm, i.e., the distance between the surface of the articular cartilage and the tidemark, which is the boundary between the non-calcified and calcified cartilage) and Osteoarthritis Research Society International (OARSI) score on safranin O (0.067%) / Fast Green (0.037%) stained sections [36]. In the secondary spongiosa of the metaphysis we quantified the adipocyte number (Fig 1d), the double labeled surfaces (dLS/BS, %) and mineral apposition rate (MAR, µm/day) on unstained sections, from which the bone formation rate (BFR/BS, µm3/µm2/day) was calculated. Osteoclast surfaces (Oc.S/BS, %) were assessed after TRAP staining.

## Skeletal muscles

Skeletal muscles from the six out of 16 animals not infused with barium sulfate were weighted using the Mettler ToLedo scale (Columbus, OH) prior to processing. The right tibialis anterior

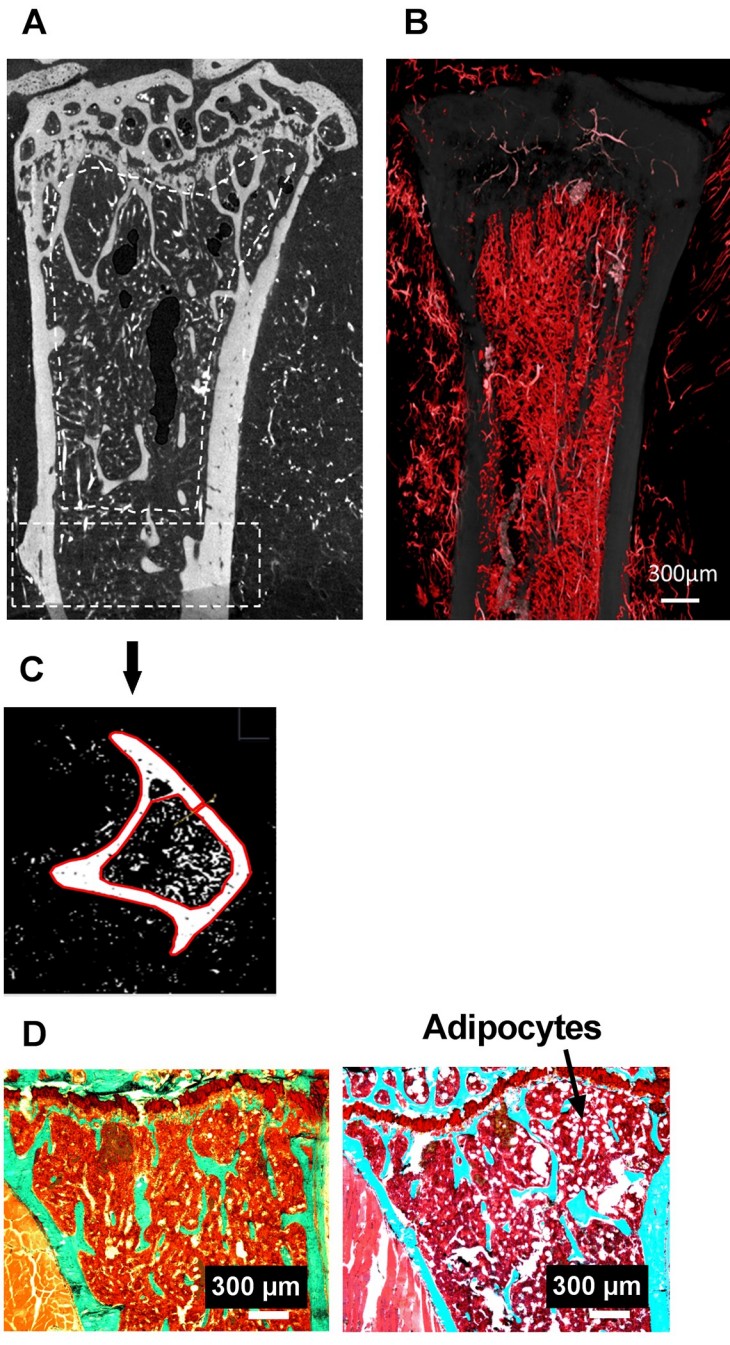

**Fig 1. Bone and vascular structural parameters measured at the right tibia using nanoCT at 3 μm resolution.** (a) metaphyseal and (c) cortical region of interest (ROI). In (a), blood vessels filled with barium sulfate appear in white, bone in grey. (b) Stack of 46 images, each 3 μm thick, processed in ImageJ with white/red color assignment to pixels corresponding to the vascular sector. (c) Transversal section in the diaphysis, bone and blood vessels appear in white after thresholding, the red contours delimit the cortical bone (d) Tibial metaphysis with a safranin O/Fast Green stain used for the quantification of adipocytes. On the left an image (bar is 500 μm) of the Ctrl 2g group and on the right an image of the OA 2g group.

and soleus muscles were rapidly included in Optimal Cutting Temperature compound (OCT, Tissue Tek®, Sakura, France) and stored at -80˚C for later immunohistochemistry (IHC) analyses. Soleus and tibialis muscle OCT blocks were cut transversally using a cryo-microtome

(Micro HM 560, USA). Hematoxylin-Eosin staining was used for fibers and cell nucleus quality checking, ATPase staining for soleus muscle fiber typing, oil red-O staining for lipid droplets, and immunohistochemistry of laminin (L9393, Sigma-Aldrich, Corp. St. Louis, MO USA) for fibers size and number quantification. Lastly, a double labeling CD31/laminin (CD31, AF3628, Goat, anti-mouse antibody, R&D Systems, 614 McKinley Place, Minneapolis, USA) was done to measure muscle vascular density (Ves. Nb/200 $\mu m^2$).

### Gene expression

The right tibia and gastrocnemius from non-infused animals were stored at -80˚C until PCR analysis. They were powdered before total RNA was extracted using Tri-reagent (Sigma Aldrich). The extraction and purification of RNA were made according to standard method. Then reverse transcriptase (BioRad standard protocol) was made to synthetize cDNAs (1µg/ 100 µL). All PCRs were performed on the BioRad CF96 Real Time System (C1000 Thermal Cycler). The cDNAs samples were diluted to a concentration of 2.5µg/mL. The primers used are in S1 Table in S1 File.

### Serum analyzes

Serum from non-infused mice was obtained after decapitation and 1 min centrifugation at 15000g (400–600 µl per mouse). We measured the concentration of TNF-α (R & D System, Quantikine HS ELISA MHSTA50), Corticosterone (Elabscience Mouse CORT (Corticosterone) Elisa Kit E-EL-MO349), Visfatin (LSBio NAMPT / Visfatin ELISA Kit LS-F4385) and Irisin (LSBio FNDC5 / Irisin ELISA Kit LS-F23848). The coefficients of variations for these measurements are less than 10%.

### Statistical analyses

Individual values and box and whisker plots showing the 25th through 75th percentiles were plotted with the central horizontal line representing the mean. Two-way ANOVA followed by a Sidak post-hoc test in case of significance ($p < 0.05$) was used to compare Ctrl and OA on one way, and 1g and 2g on the other way (GraphPad prism 7.0 software).

## Results

Number of mice in each group and individual values for all the parameters reported in this section are in S2 Table in S1 File.

Body mass was unaffected by either condition (30.8 ± 1.74 g for Ctrl 1g, 33 ± 1.73 g for OA 1g, 30.3 ± 1.85 g for Ctrl 2g and 31 ± 3.1 g for OA 2g).

### Osteoarthritis

OARSI score was significantly increased in both OA groups, compared to their respective controls. However, score remained lower than 3, 8 weeks post-surgery, indicating moderate OA (Fig 2). Osteophyte formation at the medial side of the surgical joint were present in 90% of OA 1g mice and in 80% of OA 2g mice. The articular cartilage thickness decreased by 43% in OA 1g, and by 27% in OA 2g, vs their respective controls (not shown). The subchondral bone thickness increased by 45% in OA 1g, but preserved in OA 2g, vs their respective controls (not shown). These observations indicated that hypergravity conditions alleviated OA severity.

## A

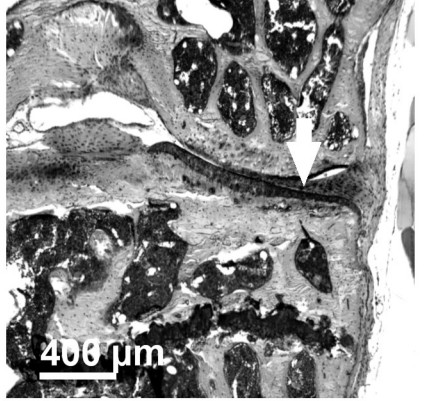 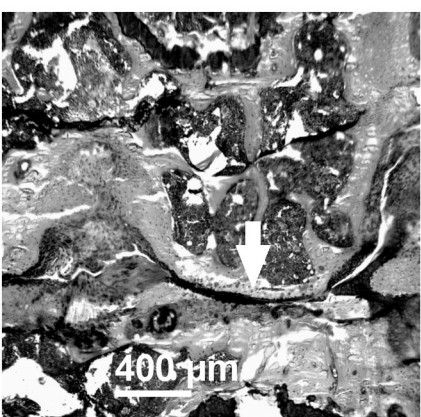

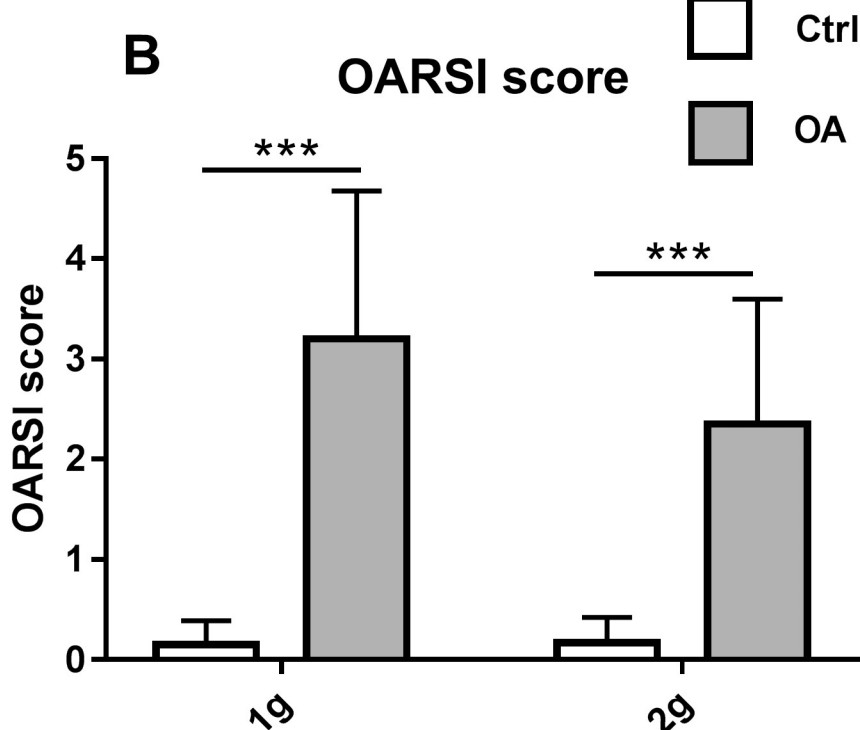

**Fig 2. OA assessment.** (a) Illustration of histological Safranin O/ Fast Green stained sections of a Ctrl 1g (left) and OA 1g (right) knee. The white arrows indicate healthy (left) and OA (right) cartilage (b) OARSI scores at 8 weeks post-operatively (mean± SD); (individual values and box and whisker plots showing the 25th through 75th percentiles boxed and the central horizontal line the mean. The data range is represented by whiskers. Measurements were performed at the medial site. *** p<0.001, 2-way ANOVA followed by a Sidak post-hoc test, sham controls (white box Ctrl, n = 10) and OA (grey box, n = 10) groups at 1g earth gravity or 2g centrifugation.

## Tibia metaphysis and diaphysis bone parameters

In OA 1g, trabecular BV/TV was 28% lower as compared to Ctrl 1g. This was associated to fewer trabeculae (-23% in Tb.N, vs. Ctrl 1g) and more active osteoclastic surfaces (+48%, vs. Ctrl 1g). Trabecular bone loss was fully prevented by hypergravity exposure in OA 2g mice (Fig 3a, 3c and 3e), likely due to the lack of stimulation of bone resorption (Fig 3d). Although MAR changes did not reach a significant p value, a small reduction was observed in OA 1g vs Ctrl 1g while a slight increase was observed in OA 2g vs Ctrl 2g. This resulted in a 18% significant increase in OA 2g vs OA 1g (Fig 3f). Cortical bone mass was unaffected by OA at 1g and at 2g, as compared to their respective controls (Fig 3b). Nevertheless, cortical thickness of OA 2g mice was 9% lower than that of OA 1g. Cortical porosity was not different among the four groups (Table 1).

Overall, our results showed that hypergravity fully prevented OA bone mass or cellular alteration only in the trabecular compartment.

Paradoxically, bone gene expression of runx2, a marker of osteoblast differentiation, was significantly increased (+150% vs. Ctrl 1g) by OA at 1g and not at 2g (Fig 4c), with no change in osteocalcin or sclerostin expression (Table 2). Similarly, mRNA expression of TNF-α, a pro-inflammatory cytokine, was higher in OA 1g mice (175% vs Ctrl 1g), an effect fully prevented by hypergravity in OA 2g mice (Fig 4e).

Surprisingly, we observed a 3.5-fold increase in bone marrow adipocyte density in tibia metaphysis in OA 2g compared to Ctrl 2g (Fig 4a), a feature not present in OA 1g. In contrast expression of CEBP-α, a marker of adipocyte differentiation was upregulated by OA at 1g (+98%, vs Ctrl 1g), whereas it did not change in OA 2g mice.

Marrow vessel density was reduced by hypergravity (-40%) in Ctrl 2g compared to Ctrl 1g. OA did not affect vessel density at 1g, whereas, it prevented hypergravity-induced decrease in vessel number in OA 2g (Fig 4b).

## Skeletal muscles

OA at 1g did not affect soleus, tibialis or gastrocnemius muscle masses (Fig 5a, 5e and 5g).

In control mice, hypergravity increased the mass of the soleus muscle (an antigravity muscle) by 30% (Fig 5a), but also the mass of tibialis (+11%, Fig 5e) and gastrocnemius (+9%, Fig 5g). In the soleus, the greater mass was associated with an enlargement of fiber area and a switch from type II to type I fiber (Fig 5b and 5c). In the soleus, hypergravity decreased vessel density (-37% in Ctrl 2g vs Ctrl 1g, Fig 5d).

Regarding soleus and tibialis anterior masses, OA 2g did not display any differences as compared to their Ctrl 2g. As compared to OA 1g, OA 2g showed an increase in soleus fiber area (+43%, vs OA 1g) and alteration in fiber type with more Type I (oxidative) and less Type II (glycolytic) fibers compared to OA 1g (Fig 5b and 5c). In addition, hypergravity in OA 2g mice fully prevented the increase in the number of tibialis fibers containing lipid droplets of OA at 1g (+48% in OA 1g vs Ctrl 1g) (Fig 5f). Tibialis vessel density or number is not affected in any conditions (Table 2).

In the gastrocnemius muscle, the increase in hypergravity-related muscle mass in Ctrl 2g was lost in OA 2g (Fig 5g).

At 1g, OA did not change expression of muscle-related gene expression (Fig 6). Hypergravity upregulated gene expression of myostatin (+102%), its antagonist follistatin (+187%) and its receptor Act RIIb (+994%), as well as that of FNDC5 (+746%) and visfatin (+196%) (Fig 6a–6e) compared to Ctrl 2g. Similarly, the expression of adipocyte-related genes also increased under hypergravity, including C-EBPα (+827%) and FAT/CD36 (+143%), (Fig 6f and 6g).

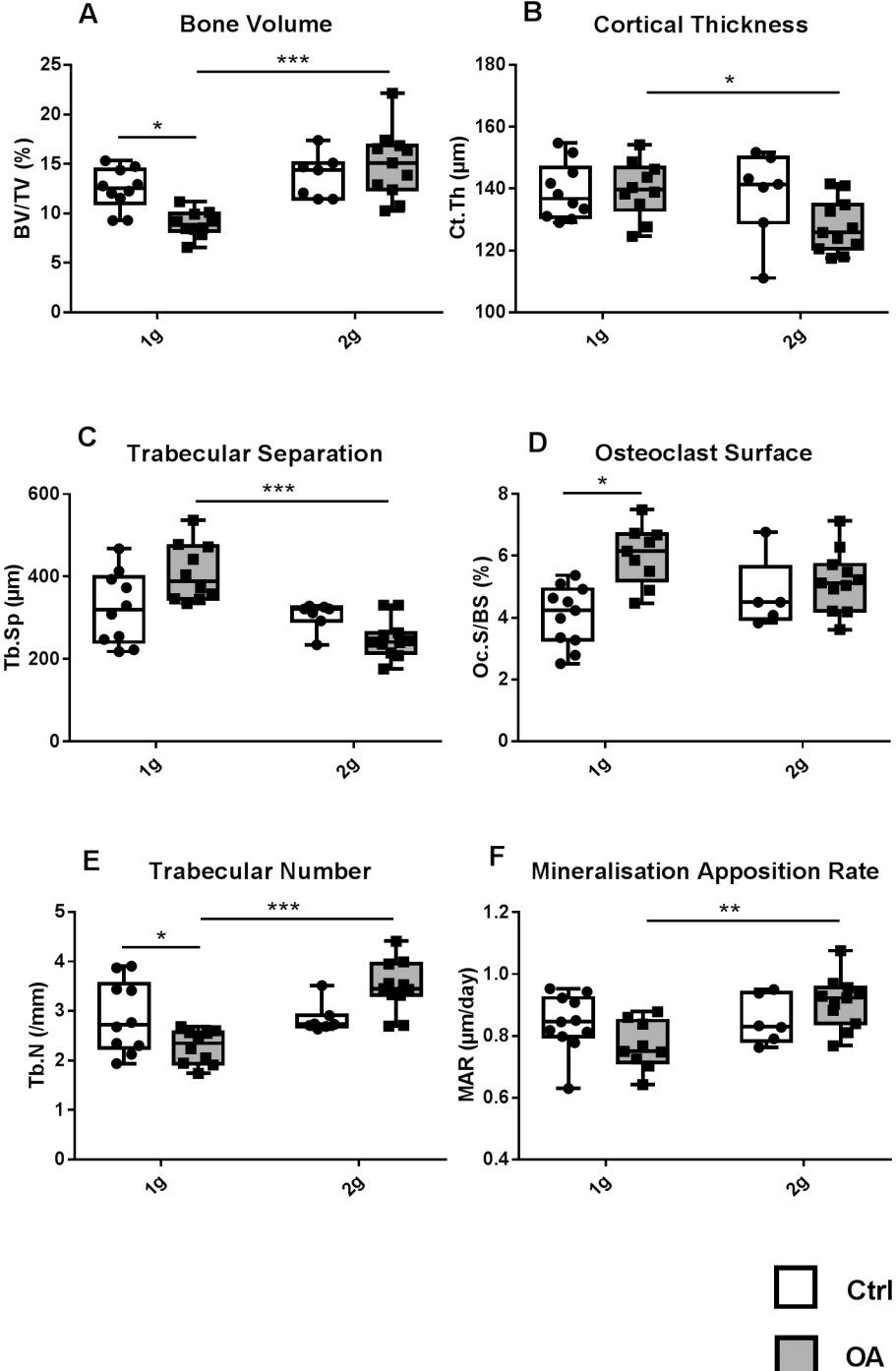

**Fig 3. Tibia, trabecular and cortical bone parameters.** Trabecular parameters in the seconday spongiosa of proximal metaphysis, i.e., bone volume (BV/TV, a) trabecular separation (Tb.Sp, c) and number (Tb.N, e) are measured in the proximal metaphysis by nano-CT. Cortical thickness (Ct.Th, b) is measured mid-diaphysis by nano-CT. Bone histomorphometry at the proximal metaphysis showing TRAP-positive osteoclast surface (Oc.S/BS, d) and mineral apposition rate (MAR, f). Individual values and box and whisker plots showing the 25th through 75th percentiles boxed and the central horizontal line the mean. The data range is represented by whiskers. $^*$ p<0.05, $^{**}$ p<0.01, $^{***}$ p<0.001, 2-way ANOVA followed by a Sidak post-hoc test, sham controls (white box Ctrl, n = 6–11) and OA (grey box, n = 10–11) groups at 1g earth gravity or 2g centrifugation.

**Table 1. Bone parameters (mean±SD) that did not display any difference between groups.**

|  | Ctrl 1g | Ctrl 2g | OA 1g | OA 2g |
|---|---|---|---|---|
| **Tb.Th, μm** | 45.3 ± 7.7 | 48.3 ± 5.6 | 39.57 ± 4.1 | 42.55 ± 5.3 |
| **Ct.Po (%)** | 0.54 ± 0.09 | 0.49 ± 0.11 | 0.57 ± 0.13 | 0.52 ± 0.11 |
| **Vascular volume/Marrow Volume (%)** | 5.4 ± 1.2 | 3.1 ± 0.8 | 4.8 ± 1.5 | 5.3 ± 0.7 |
| **Osteocalcin (relative expression, AU)** | 1.0 ± 0.6 | 0.29 ± 0.20 | 0.85 ± 0.50 | 0.72 ± 0.60 |
| **SOST (relative expression, AU)** | 1.0 ± 0.5 | 0.84 ± 1.1 | 1.9 ± 1.8 | 1.4± 1.5 |
| **PPAR-gamma (relative expression, AU)** | 1.0± 0.4 | 0.6 ± 0.2 | 1.0 ± 0.5 | 0.5 ± 0.1 |
| **BFR/BS (μm³/μm²/day)** | 18.7 ± 6.8 | 21.7 ± 4.8 | 23.7 ± 4.7 | 23.2 ± 3.9 |

Tb.Th, Trabecular Thickness; Ct.Po, Cortical Porosity; BFR/BS, Bone Formation Rate; Ctrl, control; OA, osteoarthritis.

These increases were maintained in OA 2g as compared to OA 1g, even if they were of smaller magnitude only for FNDC5.

## Serum

In control groups, eight weeks of hypergravity did not alter the serum levels of corticosterone or inflammation. However, corticosterone concentrations were higher in OA 2g than OA 1g (+102%, Fig 7a). In addition, while OA tended to increase TNF-α serum levels at 1g (vs Ctrl 1g, p = 0.08), hypergravity fully prevented this increase in OA mice. Hence, serum concentrations of TNF-α were significantly reduced in OA 2g compared to OA 1g (-77%) (Fig 7b). The circulating levels of visfatin and irisin were not modified (Fig 7c and 7d).

## Discussion

Following the destabilization of the medial meniscus, its medial displacement occurs, and weight is transmitted across a small area on the medial side, leading to increased local mechanical stress. Since the mouse knee is usually flexed, this results in greater stress on the posterior medial femoral condyle and medial tibial plateau [34]. After 8 weeks, we, as others [37, 38] observed a moderate OA, characterized by a thinning of the tibia articular cartilage, subchondral sclerosis and the presence of osteophytes. At the end of the experiment, body mass was unaltered in both groups of centrifuged mice, in accordance with our previous observation in younger mice [31].

Our first goal was to evaluate musculoskeletal alteration in OA. At the tibia level, OA mice displayed site and compartment specific responses. At the midshaft, cortical bone was preserved. However, at the proximal metaphysis, trabecular bone volume decreased due to reduced trabecular number, increased osteoclastic resorption, and increased local TNFα gene expression, a potent proinflammatory cytokine playing a crucial role in stimulating osteoclastogenesis [39]. Of note, serum level of TNFα showed a tendency to increase suggesting the emergence of a low-grade systemic inflammation. At the muscle limb level, no change was noted for functional test (Kondziela test [40], S2 Table in S1 File) or muscle masses. However, trend towards decrease in soleus fiber cross sectional area and a significant increase in the proportion of tibialis fibers containing lipid droplets were observed. This intramuscular fat invasion likely reflects a decline in lipid turnover [41]. Unexpectedly, we found no change in adipocyte-related gene expression in muscles suggesting that eight-week after DMM surgery, gene expression normalized. Indeed, earlier observation [41] showed that a shift in metabolism, characterized by a loss in efficiency of lipid metabolism, occurs in advance of the onset of sarcopenia being thus causative event that contributes to muscle vulnerability. Overall, our

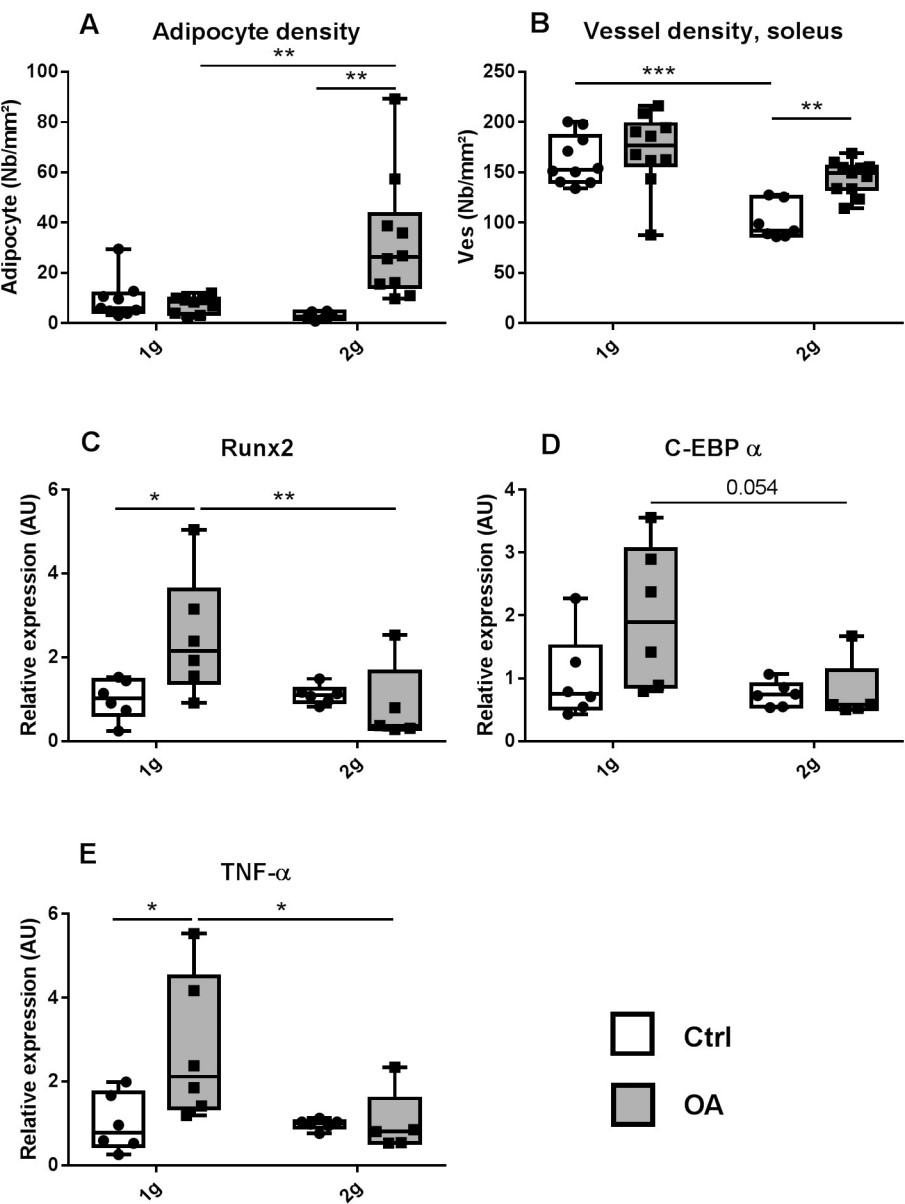

**Fig 4.** Tibia metaphysis, marrow adipocyte number (a) and vessel density (b) (white box Ctrl, n = 10) and OA (grey box, n = 10) groups at 1g earth gravity or 2g centrifugation. Expression of genes involved in osteoblastogenesis Runx2 (c), in adipogenesis C-EBPα (d) and in inflammation TNF-α (e). All data were obtained by quantitative real PCR analysis of RNA and HPRT as the relative control. Individual values and box and whisker plots showing the 25th through 75th percentiles boxed and the central horizontal line the mean. The data range is represented by whiskers. * p<0.05, ** p<0.01, 2-way ANOVA followed by a Sidak post-hoc test, sham controls (white box Ctrl, n = 6) and OA (grey box, n = 6) groups at 1g Earth gravity or 2g centrifugation.

findings demonstrated that knee OA induced effects distant from the joint, characterized by metaphyseal bone deterioration and premise of sarcopenia. At this stage, the influence of OA in our model appears to be limited to the cancellous bone tissue but it is also possible that changes in cortical bone occur much more slowly and take longer to develop than changes in the more metabolically active cancellous bone. Different regulatory mechanisms driven by the differential gene transcription between cancellous and cortical bone might also account for their different responses, as shown in loaded C57BL/6J tibiae [42].

**Table 2. Muscle parameters (mean±SD) that did not display any difference between groups.**

|  | Ctrl 1g | Ctrl 2g | OA 1g | OA 2g |
|---|---|---|---|---|
| Vessel Number per fibers, soleus | 3.29 ± 0.40 | 2.45 ± 0.20 | 2.77 ± 0.20 | 2.50 ± 0.40 |
| PPAR-gamma (relative expression, AU), gastrocnemius | 1.00 ± 0.72 | 1.45 ± 0.20 | 1.27 ± 0.71 | 1.47 ± 0.90 |
| Area fibre (μm$^2$), tibialis | 2004.5 ± 134.1 | 1945.5 ± 202.5 | 1913.8 ± 296.7 | 1952.1 ± 241.1 |
| Vessel density tibialis (/200μm$^2$) | 27.9 ± 4.3 | 32.9 ± 9.6 | 33.5 ± 7.5 | 31.4 ± 13.2 |
| Vessel Number per fibers, tibialis | 2.2 ± 0.4 | 2.15 ± 0.5 | 2.64 ± 0.6 | 2.65 ± 0.5 |

We then tested whether hypergravity was able to maintain or even strengthen bone and muscle. In non-OA mice, no alteration was produced by 2g exposure at the bone structural or bone remodeling level. This lack of effect contrasts with our previous findings using younger mice (7-weeks of age when sacrificed) exposed to 2g for a shorter time (3 weeks) as compared to mice of the current study (24-weeks of age at sacrifice, 8 weeks at 2g). In these young mice, we observed an increase in cancellous bone volume at the distal femur along with reduced osteoclastic surfaces [31]. In OA-mice however, hypergravity did show an effect since it partially protected from osteoarthritis severity and associated bone degradation. Indeed, trabecular bone parameters are reinforced at proximal tibia with stimulation of bone formation and prevention of OA-induced osteoclastic bone resorption. However, cortical thickness decreased as compared to OA 1g. AO associated with hypergravity therefore appears to weaken the cortical bone but strengthen the trabecular bone. It would be necessary to check if these changes alter the overall biomechanical properties. The increase in bone marrow adipocytes in OA 2g disagrees with the decrease in the expression of C-EBPα factor. It is possible that the appearance of adipocytes followed an early and transient increase in these factors that then slowed down. However, at this stage, we have no explanation for the fact that OA 2g have a marrow fat invasion not seen in the other groups.

At the skeletal muscle level, 2g hypergravity is anabolic. In intact non-OA mice, calf muscle hypertrophied by 30% in the antigravity soleus muscle and a little more than 10% in tibialis and gastrocnemius muscles. Further, larger fibers were seen in soleus. Thus, in non-OA mice, hypergravity has characteristics similar to those found in resistance exercises. Contrary to Ctrl 2g, OA 2g mice, did not show any increase in muscle mass, since soleus, tibialis and gastrocnemius were similar to those at 1g. However, in both Ctrl and OA, 2g hypergravity induced phenotypical changes with a switch from type-II to type-I fiber in the soleus. These findings suggest that hypergravity stimulates the oxidative capacity of type-I fibers, as in endurance training that typically results in an overall shift away from type-II expressing fibers to a more oxidative phenotype expressing type-I muscle fibers in both human and rodents [43–45]. In rats, it has also been observed that both exercise (voluntary wheel running) and chronic 2g caused an increase in the slow myosin heavy chain in soleus muscle [46], suggesting that loading is a primary stimulus for this shift. Furthermore, hypergravity blocked lipid deposits in the OA 2g tibialis. Thus, if being osteoarthritic prevents muscle mass gain, the muscle morphology and typology characteristics of 2g mice were retained.

At 2g, the expressions of some adipocyte-related markers, C-EBPα and FAT/CD36, but not all (PPARγ), were increased in the gastrocnemius. Nevertheless, these molecular changes are not associated with signs of lipid invasion (no increase in lipid droplets in the tibialis in CTR 2g and decrease in OA 2g as compared to OA 1g). If we have no clear explanation for C-EBPα, a late marker of adipocyte differentiation, the increased expression of FAT/CD36 might reflect an increased need for fatty acid delivery and oxidation, as seen during physical exercise [47]. Investigation of muscle molecular pathways indicated that hypergravity dramatically

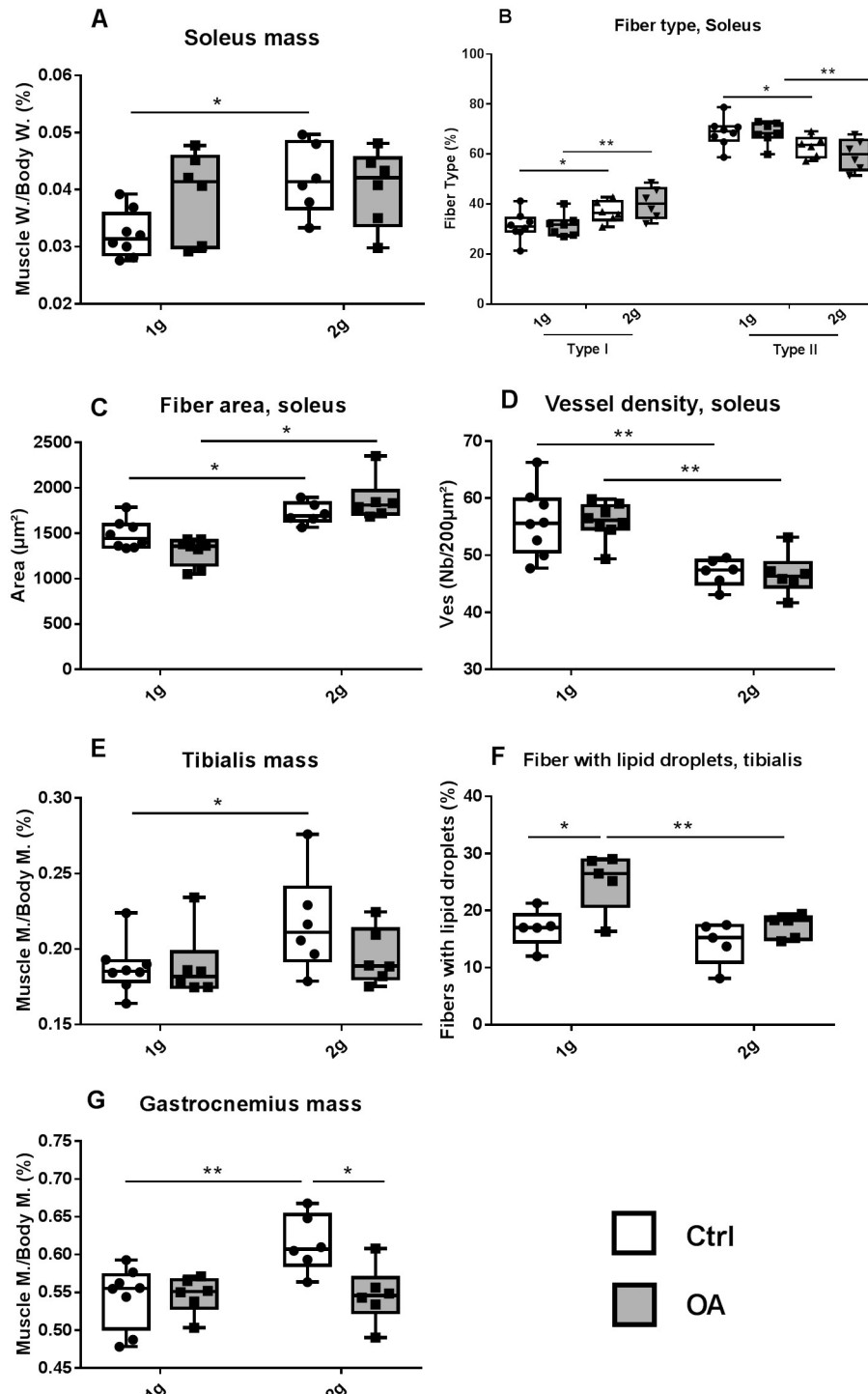

**Fig 5. Hindlimb muscles.** Relative muscle mass normalized by body mass of the soleus (a), tibialis (e), and gastrocnemius (g); Proportion of fiber type I and type II in the soleus (b), area of muscle fibers in the soleus (c) and vessel density (double CD31a and laminin positive blood vessels) in the soleus (d). Percent fibers containing lipid droplets in oil-O red staining sections of the tibialis (f). Individual values and box and whisker plots showing the 25th through 75th percentiles boxed and the central horizontal line the mean score. The data range is represented by whiskers. * $p < 0.05$, ** $p < 0.01$, 2-way ANOVA followed by a Sidak post-hoc test, sham controls (white box Ctrl, n = 6–8) and OA (grey box, n = 6) groups at 1g earth gravity or 2g centrifugation.

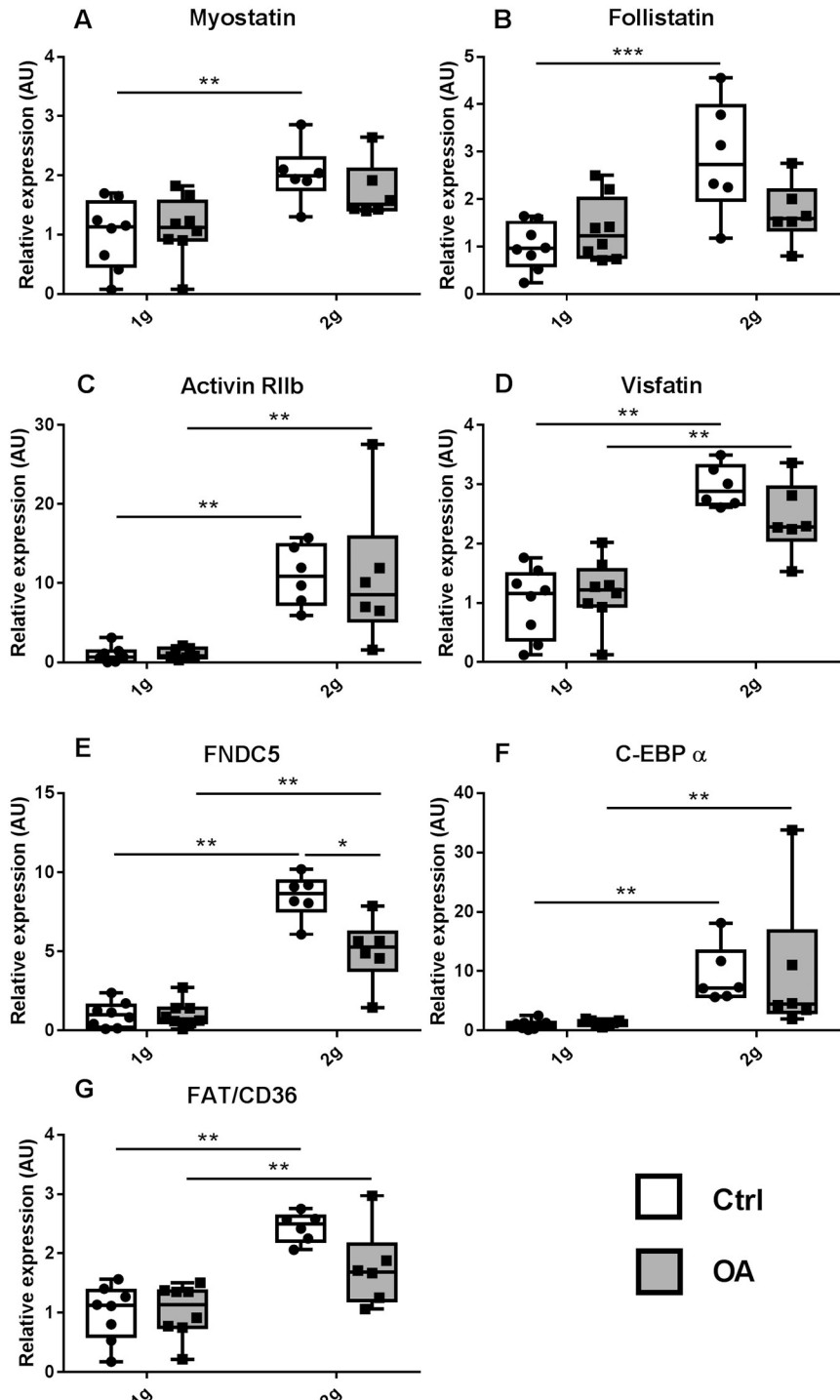

**Fig 6. Expression of genes involved in gastrocnemius muscle metabolism, myostatin (a), its antagonist follistatin (b), its receptor activin RIIb (c), visfatin (d), FNDC5 (e) and in fat metabolism C-EBPα (f) and FAT/CD36 (g).** All expression data were obtained by quantitative real PCR analysis of RNA and HPRT as the relative control. Individual values and box and whisker plots showing the 25th through 75th percentiles boxed and the central horizontal line the mean score. The data range is represented by whiskers. * p<0.05, ** p<0.01, *** p<0.001, 2-way ANOVA followed by a Sidak post-hoc test, sham controls (white box Ctrl, n = 6–8) and OA (grey box, n = 6–8) groups at 1g earth gravity or 2g centrifugation.

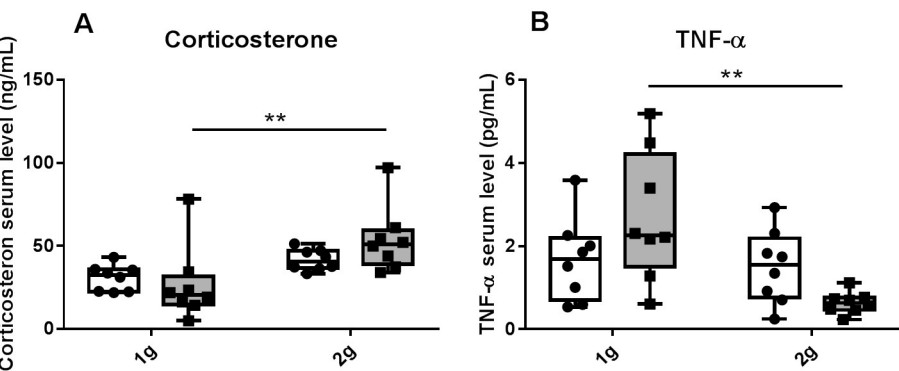

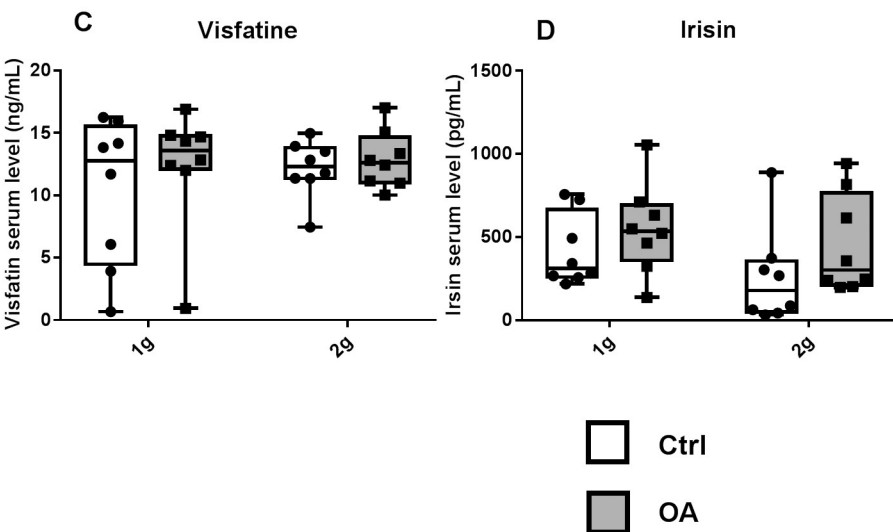

**Fig 7. Serum level of stress hormone, inflammatory markers or adipo-myokines: corticosterone (a), TNF-α (b), visfatine (c) and irisin (d).** Individual values and box and whisker plots showing the 25th through 75th percentiles boxed and the central horizontal line the mean score. The data range is represented by whiskers. ** $p < 0.01$, 2-way ANOVA followed by a Sidak post-hoc test, sham controls (white box Ctrl, n = 8) and OA (grey box, n = 8) groups at 1g earth gravity or 2g centrifugation.

stimulated the expression of both catabolic (myostatin and its receptor activin RIIb, visfatine), and anabolic (FNDC5, follistatin) signaling mediators in Ctrl mice suggesting that hypergravity stimulated muscle remodeling. In OA mice, catabolic genes were similarly stimulated by 2g but the anabolic genes FNDC5 and follistatin were stimulated to a lesser extent than in non-AO 2g mice. This might indicate an imbalance between anabolic and catabolic pathways and explain why no gain in muscle mass was seen in OA 2g. Overall, the results observed in the 2g groups (CONT and OA) are an additional argument showing the parallel between physical exercise and exposure to hypergravity. Of note, these results obtained in the gastrocnemius muscle (movement muscle) might not be extrapolated to other muscles, particularly postural muscles such as soleus, which are constantly overloaded in this model [48]. It is also possible that muscle fibers activated metabolic pathways in response to the increase in circulating corticosterone, which might be a confounding factor in OA 2g. However, in Ctrl 2g the

corticosterone only showed a tendency to rise that could be associated to increased physical demand [49].

In response to resistive exercise training, increased capillary density and vascularization are evidenced, due to increased oxygen and nutrients delivery [46]. However, contrary to our expectations [31], the vascular density decreased in the soleus muscle as well as in the bone marrow, in 2g mice. Therefore, not all hypergravity-related features mimic those of resistive exercise. Blood flow redistribution under increased gravitational forces is an accepted although understudied feature. Such redistribution has been reported in humans exposed to hypergravity in supine postures where it influences lung regional distribution of both blood flow and ventilation [50]. Centrifugation [2 to 3 g in direction head-pelvis in prone position] has been shown to improve peripheral circulation in patients with obliterative atherosclerosis of lower extremity arteries [51].

The first limitation of this study was not directly measuring muscle strength. Despite this, the non-specific Kondziela test [40] did not show any difference between groups, nor the physical activity of non-OA and OA mice in the centrifuge. We tested running wheel capacity in the centrifuge in 3 extra mice in 2g Ctrl and 2g OA (S3 Table in S1 File) and found no differences suggesting that 2g OA did not experience pain to a point that it had affected running capacities. The second limitation was not being performing biomechanical testing to assess material consequence of reinforcement of trabecular bone associated to thinner cortex in the tibia.

In conclusion, we demonstrated the effects of OA on the musculoskeletal system in a mouse model. We also showed that gravitational strength, mimic some, not all, features of resistance physical exercise in non-OA mice. In OA mice, 2g hypergravity has mixed effects with positive outcomes for trabecular bone and muscle typology, but negative effects for cortical bone. Modalities of application (level of g, time and frequency of application) can be challenged in order to prevent musculoskeletal effects of osteoarthritis.

## Supporting information

**S1 File.**
(DOCX)

## Acknowledgments

We would like to gratefully acknowledge the PLEXAN animal facility for animal breeding and care.

## Author Contributions

**Conceptualization:** Marie-Hélène Lafage-Proust, Laurence Vico.

**Data curation:** Benoit Dechaumet, Damien Cleret, Marie-Thérèse Linossier, Stéphanie Chanon, Etienne Lefai, Marie-Hélène Lafage-Proust.

**Formal analysis:** Benoit Dechaumet, Marie-Thérèse Linossier, Norbert Laroche.

**Funding acquisition:** Marie-Hélène Lafage-Proust.

**Investigation:** Benoit Dechaumet, Damien Cleret, Marie-Thérèse Linossier, Arnaud Vanden-Bossche.

**Methodology:** Benoit Dechaumet, Damien Cleret, Marie-Thérèse Linossier, Arnaud Vanden-Bossche, Stéphanie Chanon, Norbert Laroche.

**Project administration:** Marie-Hélène Lafage-Proust, Laurence Vico.

**Visualization:** Damien Cleret, Etienne Lefai, Laurence Vico.

**Writing – original draft:** Benoit Dechaumet.

**Writing – review & editing:** Marie-Hélène Lafage-Proust, Laurence Vico.

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
