## [Decision Letter · Decision Letter 0]

2 Sep 2020

PONE-D-20-19501

Hypergravity as a Gravitational Therapy Mitigates the Effects of Knee Osteoarthritis on the Musculoskeletal System in a murine model

PLOS ONE

Dear Dr. Vico:

Thank you for submitting your manuscript to PLOS ONE. After careful consideration, we feel that it has merit but does not fully meet PLOS ONE’s publication criteria as it currently stands. Therefore, we invite you to submit a revised version of the manuscript that addresses the points raised during the review process by an expertise reviewer and editorial comments as noted below.

We look forward to receiving your revised manuscript.

Kind regards,

Dr. Sakamuri V. Reddy

Academic Editor

PLOS ONE

Journal Requirements:

2.We noticed you have some minor occurrence of overlapping text with previous publications, which needs to be addressed. In your revision ensure you cite all your sources (including your own works), and quote or rephrase any duplicated text outside the methods section. Further consideration is dependent on these concerns being addressed.

Specific examples of the overlap include:

*Introduction: sentence beginning "Patients experience pain... Daily activities...." - this overlaps considerably with: https://academic.oup.com/ptj/article/98/7/560/4998860

*Introduction: "Bone and muscle closely interact..." - this overlaps with: https://link.springer.com/article/10.1007/s00198-017-4151-8

*Introduction: "In others, exercise interventions...." - this overlaps with: https://www.oarsijournal.com/article/S1063-4584(18)30018-9/fulltext

These are just some brief examples, there are other areas of text where the sentences/phrases appear in previously published works, but it's very important all submitted text is your own original writing.

Additionally, please provide complete information about all efforts to minimize suffering peri- and post-operatively, including full analgesic regimen, supportive care and so forth. Please also state the mortality rate of the animals (if any) due to the experimental procedures.

Additional Editor Comments (if provided):

The authors show hypergravity in osteoarthritis mice (OA) has positive effects on trabecular bone and skeletal muscle typology, however negative effects on cortical bone. Specific comments to improve the manuscript are as follow: Methods-(p.5; line 109) verify the name correctly for mouse strain used ie., C57BL/6J. They can show Table 1: List of primers used for body mass index, bone and muscle genes in supplemental. Results-they can include a rationale to study bone and skeletal muscle parameters in osteoarthritis conditions. Figures-improve quality of all the figures and labeling. Supplemental material-please clarify taking a picture by Video “centri 2014.avi” or video display properly.

Reviewers' comments:

Reviewer's Responses to Questions

**Comments to the Author**

1. Is the manuscript technically sound, and do the data support the conclusions?

Reviewer #1: Yes

2. Has the statistical analysis been performed appropriately and rigorously? 

Reviewer #1: Yes

3. Have the authors made all data underlying the findings in their manuscript fully available?

Reviewer #1: No

4. Is the manuscript presented in an intelligible fashion and written in standard English?

Reviewer #1: Yes

5. Review Comments to the Author

Reviewer #1: 1.The study presents the results of original research.

2. Results reported have not been published elsewhere.

3. Experiments, statistics, and other analyses are performed to a high technical standard and are described in sufficient detail.

4. Conclusions are presented in an appropriate fashion and are supported by the data.

5. The article is presented in an intelligible fashion and is written in standard English.

6. The research meets all applicable standards for the ethics of experimentation and research integrity.

7. The article adheres to appropriate reporting guidelines and community standards for data availability.

The manuscript by Dechaumet and colleagues entitled “Hypergravity as a Gravitational Therapy Mitigates the Effects of Knee Osteoarthritis on the Musculoskeletal System in a murine model” is an interesting paper exploring the impact on and possible application of chronic centrifugation in osteoarthritis.

It is an interesting and well-prepared and executed study exploring the effects based on both histological and molecular data.

General comments:

Experiment set-up in centrifuge. The referred to S1 video was not accessible for review). The centrifuge was started and stopped a few times during the study. Also, in addition to data from reference 31: Please identify what the spin-up and spin-down time was for this experiment.

Results:

Body mass was unaffected by either condition: This is a surprising observation and should be addressed and discussed some more. In the work from ref 46 (Fuller et al.) In that paper from Fuller et al. there is a clear reduced body mass also at 2g as well as a difference in mass with wheel running while no changes are seen in this study. But also in Picquet et al (10.1152/ajpregu.00643.2001) there was a body weight decrease of 40% after 100 days at 2g, so quite a comparable set-up and, if I am not mistaken, performed in the same centrifuge. But seen in many other studies also in other species e.g. rabbit (see DOI: 10.1152/jappl.1978.45.1.51).

Regarding body mass data: Please provide start values for this be as “Supplemental data”.

In figure 6 there is an increase in adipocyte-related gene expression of C-EPBa and FAT/CD36 in Ctr 2g compared to Ctr 1g animals. CD36 has been linked to phenotypic features of the metabolic syndrome including insulin resistance and dyslipidemia, a phenomenon related to altered gravity. How does data from this study related to previous studies where hypergravity decreases fat related parameters; see e.g. Warren et al. J Appl Physiol 90: 606–614, 2001. Using lean and obese rats? Please elaborate in discussion.

Minor comments:

- L197: CV? Please explain.

- L264: ‘since”?

- L282: no changes found in functional tests (data not shown). However, I would propose to include the data as Supplemental material for future reference. Same for run wheel data (L341)

- L317/L337: gravity should be with lower case ‘g’

- Fig. 1: Although quite common maybe good to explain ‘ROI’.

- Fig. 1: In general, the images are of poor resolution which might be because of the journal conversion to pdf. Presume the originals are better

- Fog. 1: Difference between gray and white (bone mineral / blood vessel) not very clear in a and c. c seems to be over exposed. Cortex nearly all white (and yellow?).

- Fig 1: “of the Ctrl group”. What control group? Ctrl-1g or AO-1g? Please clarify

- Fig. 2: “O/ Fast Green” should be “safranin O/Fast Green”

- Fig. 2: Also, here identify the “Ctrl” better.

- Fig. 4: Earth with capital ‘E’

6. PLOS authors have the option to publish the peer review history of their article (what does this mean?). If published, this will include your full peer review and any attached files.

Reviewer #1: No

---

## [Author Response · Author response to Decision Letter 0]

13 Nov 2020

2.We noticed you have some minor occurrence of overlapping text with previous publications, which needs to be addressed. In your revision ensure you cite all your sources (including your own works), and quote or rephrase any duplicated text outside the methods section. Further consideration is dependent on these concerns being addressed.

Specific examples of the overlap include: (they have been reformulated)

*Introduction: sentence beginning "Patients experience pain... Daily activities...." - this overlaps considerably with: https://academic.oup.com/ptj/article/98/7/560/4998860

*Introduction: "Bone and muscle closely interact..." - this overlaps with: https://link.springer.com/article/10.1007/s00198-017-4151-8

*Introduction: "In others, exercise interventions...." - this overlaps with: https://www.oarsijournal.com/article/S1063-4584(18)30018-9/fulltext

These are just some brief examples, there are other areas of text where the sentences/phrases appear in previously published works, but it's very important all submitted text is your own original writing.

Additionally, please provide complete information about all efforts to minimize suffering peri- and post-operatively, including full analgesic regimen, supportive care and so forth. Please also state the mortality rate of the animals (if any) due to the experimental procedures. No mortality occurred. To minimize suffering peri- and post-operatively, meloxicam (0.5mg/ml) a non-steroidal anti-inflammatory drug, was given in drinking water 48h before and 48h after surgery. This was added in Material and Method section, paragraph “Surgical induction of osteoarthritis“ Page 5-6 Line 129-131 in the 'Revised Manuscript with Track Changes'

This has been done

Additional Editor Comments (if provided):

The authors show hypergravity in osteoarthritis mice (OA) has positive effects on trabecular bone and skeletal muscle typology, however negative effects on cortical bone. Specific comments to improve the manuscript are as follow: Methods-(p.5; line 109) verify the name correctly for mouse strain used ie., C57BL/6J. Thank you it has been corrected

They can show Table 1: List of primers used for body mass index, bone and muscle genes in supplemental. This has been moved in Supplementary Material (S1 Table)

Results-they can include a rationale to study bone and skeletal muscle parameters in osteoarthritis conditions. The rational is in paragraph 2 of the INTRODUCTION section (line 59-68 'Revised Manuscript with Track Changes') beginning with “Overall, these studies have suggested that sarcopenia and osteopenia/osteoporosis - or osteosarcopenia - are two conditions that might be associated with OA”.

Figures-improve quality of all the figures and labeling. All the figures have been redone to meet the resolution required

Supplemental material-please clarify taking a picture by Video “centri 2014.avi” or video display properly. This has been corrected, sorry for the disagreement

Comments to the Author

1. Is the manuscript technically sound, and do the data support the conclusions?

Reviewer #1: Yes

2. Has the statistical analysis been performed appropriately and rigorously? 

Reviewer #1: Yes

3. Have the authors made all data underlying the findings in their manuscript fully available?

Reviewer #1: the individual data are all made available in S2 Table ________________________________________

4. Is the manuscript presented in an intelligible fashion and written in standard English?

Reviewer #1: Yes

5. Review Comments to the Author

Reviewer #1: 1. The study presents the results of original research.

2. Results reported have not been published elsewhere.

3. Experiments, statistics, and other analyses are performed to a high technical standard and are described in sufficient detail.

4. Conclusions are presented in an appropriate fashion and are supported by the data.

5. The article is presented in an intelligible fashion and is written in standard English.

6. The research meets all applicable standards for the ethics of experimentation and research integrity.

7. The article adheres to appropriate reporting guidelines and community standards for data availability.

The manuscript by Dechaumet and colleagues entitled “Hypergravity as a Gravitational Therapy Mitigates the Effects of Knee Osteoarthritis on the Musculoskeletal System in a murine model” is an interesting paper exploring the impact on and possible application of chronic centrifugation in osteoarthritis.

It is an interesting and well-prepared and executed study exploring the effects based on both histological and molecular data.

General comments:

Experiment set-up in centrifuge. The referred to S1 video was not accessible for review). The centrifuge was started and stopped a few times during the study. Also, in addition to data from reference 31: Please identify what the spin-up and spin-down time was for this experiment. The video is now readable, sorry for the disagreement. The duration of both spin-up and spin-down is of 40 sec. (added in Material and Method section, paragraph “Hypergravity exposure”, page 6, line 141 'Revised Manuscript with Track Changes').

Results:

Body mass was unaffected by either condition: This is a surprising observation and should be addressed and discussed some more. In the work from ref 46 (Fuller et al.) In that paper from Fuller et al. there is a clear reduced body mass also at 2g as well as a difference in mass with wheel running while no changes are seen in this study. But also in Picquet et al (10.1152/ajpregu.00643.2001) there was a body weight decrease of 40% after 100 days at 2g, so quite a comparable set-up and, if I am not mistaken, performed in the same centrifuge. But seen in many other studies also in other species e.g. rabbit (see DOI: 10.1152/jappl.1978.45.1.51). Thank you for pointing these additional references. In these studies, other species were used, rats or rabbit, which might explain the differences. In the two first studies, rats still in their growing phase were used. Thus, it is possible that hypergravity affected body weight through growth inhibition. Further, in Fuller et al. study, centrifugation was interrupted twice weekly for about 15-to 20-min periods for animal husbandry, thus more often than in our study. Another parameter is the fact that rats were individually housed; this might affect their energy metabolism and by consequence their musculoskeletal system as showed in rodents (Martin et al., 2019 doi: 10.1530/EC-19-0359). We previously observed (ref 31, Gnyubkin et al., 2015) that even in growing mice, we do observe a decreased body mass at 3g (as also found in Kawao et al., 2016 doi: 10.14814/phy2.12979), but not at 2g, after 21-day of continuous hypergravity. In ref 30 (Bojados and Jamon, 2011) it is reported that C57BL/6j mice centrifuged at 2 g from embryonic to 10 or 30 postnatal days do loose body mass while those submitted to 2g after birth did not. In the present study we confirmed, in mature mice, that body mass is unaltered at 2g. 

Regarding body mass data: Please provide start values for this be as “Supplemental data”. They have been provided

In figure 6 there is an increase in adipocyte-related gene expression of C-EPBa and FAT/CD36 in Ctr 2g compared to Ctr 1g animals. CD36 has been linked to phenotypic features of the metabolic syndrome including insulin resistance and dyslipidemia, a phenomenon related to altered gravity. How does data from this study related to previous studies where hypergravity decreases fat related parameters; see e.g. Warren et al. J Appl Physiol 90: 606–614, 2001. Using lean and obese rats? Please elaborate in discussion.

Thank you for pointing out this result that we did not comment in the discussion section. We have now added (page 15, lines 331-338 'Revised Manuscript with Track Changes'): “At 2g, the expressions of some adipocyte-related markers, C-EBPα and FAT/CD36, but not all (PPARγ), were increased in the gastrocnemius. Nevertheless, these molecular changes are not associated with signs of lipid invasion (no increase in lipid droplets in the tibialis in CTR 2g and decrease in OA 2g as compared to OA 1g). If we have no clear explanation for C-EBPα, a late marker of adipocyte differentiation, the increased expression of FAT/CD36 might reflect an increased need for fatty acid delivery and oxidation, as seen during physical exercise (ref 47)”.

Minor comments:

- L197: CV? Please explain. Coefficient of variations, corrected

- L264: ‘since”? “since” has been deleted

- L282: no changes found in functional tests (data not shown). However, I would propose to include the data as Supplemental material for future reference. These data have been added in the supplementary material (S2 Table, column AH “Kondziela score”)

Same for run wheel data (L341) Here we have fractional data in only 3 groups

- L317/L337: gravity should be with lower case ‘g’ done thank you

- Fig. 1: Although quite common maybe good to explain ‘ROI’. cortical region of interest (ROI), corrected

- Fig. 1: In general, the images are of poor resolution which might be because of the journal conversion to pdf. Presume the originals are better We have now uploaded better quality figures

- Fog. 1: Difference between gray and white (bone mineral / blood vessel) not very clear in a and c. c seems to be over exposed. Cortex nearly all white (and yellow?). 

The legend of Figure 1 has been altered accordingly, in particular Fig 1b and c. Please see below with alterations in red

“Bone and vascular structural parameters measured at the right tibia using nanoCT at 3 µm resolution. (a) metaphyseal and (c) cortical region of interest (ROI). In (a), blood vessels filled with barium sulfate appear in white, bone in grey. (b) Stack of 46 images, each 3 μm thick, processed in ImageJ with white/red color assignment to pixels corresponding to the vascular sector. (c) Transversal section in the diaphysis, bone and blood vessels appear in white after thresholding, the red contours delimit the cortical bone (d) Tibial metaphysis with a safranin O/Fast Green stain used for the quantification of adipocytes. On the left an image (bar is 500 µm) of the Ctrl 2g group and on the right an image of the OA 2g group. “

- Fig 1: “of the Ctrl group”. What control group? Ctrl-1g or AO-1g? Please clarify; It has been clarified: Ctrl-2g or AO-2g

- Fig. 2: “O/ Fast Green” should be “safranin O/Fast Green”, yes thank you corrected

- Fig. 2: Also, here identify the “Ctrl” better. We included: Ctrl 1g (left) and OA 1g (right) knee

- Fig. 4: Earth with capital ‘E’: alteration made

6. PLOS authors have the option to publish the peer review history of their article (what does this mean?). If published, this will include your full peer review and any attached files.

Do you want your identity to be public for this peer review? For information about this choice, including consent withdrawal, please see our Privacy Policy.

Reviewer #1: No

---

## [Editor Report · Decision Letter 1]

16 Nov 2020

Hypergravity as a Gravitational Therapy Mitigates the Effects of Knee Osteoarthritis on the Musculoskeletal System in a murine model

PONE-D-20-19501R1

Dear Dr. VICO,

We’re pleased to inform you that your manuscript has been judged scientifically suitable for publication and will be formally accepted for publication once it meets all outstanding technical requirements.

Kind regards,

Dr. Sakamuri V. Reddy

Academic Editor

PLOS ONE
---

## [Editor Report · Acceptance letter]

19 Nov 2020

PONE-D-20-19501R1 

Hypergravity as a Gravitational Therapy Mitigates the Effects of Knee Osteoarthritis on the Musculoskeletal System in a murine model 

Dear Dr. VICO:

I'm pleased to inform you that your manuscript has been deemed suitable for publication in PLOS ONE. Congratulations! Your manuscript is now with our production department. 

Kind regards, 

on behalf of

Dr. Sakamuri V. Reddy 

Academic Editor

PLOS ONE